# In Vitro Antioxidant and Antifungal Activities of Four Essential Oils and Their Major Compounds against Post-Harvest Fungi Associated with Chickpea in Storage

**DOI:** 10.3390/plants12203587

**Published:** 2023-10-16

**Authors:** Lamyae Et-tazy, Abdeslam Lamiri, Laila Satia, Mohamed Essahli, Sanae Krimi Bencheqroun

**Affiliations:** 1Applied Chemistry and Environment Laboratory, Faculty of Sciences and Techniques, University Hassan First, BP. 577, Settat 26000, Morocco; l.et-tazy@uhp.ac.ma (L.E.-t.); abdeslam.lamiri@uhp.ac.ma (A.L.); mohamed.essahli@uhp.ac.ma (M.E.); 2Plant Protection Laboratory, Regional Center of Agricultural Research of Settat, National Institute of Agricultural Research, Avenue Ennasr, BP. 415 Rabat Principal, Rabat 10090, Morocco; laila.satia@inra.ma

**Keywords:** essential oils, natural alternatives, antioxidant activity, antifungal activity

## Abstract

The antifungal and antioxidant properties of essential oils (EOs) derived from four plants were assessed in vitro: *Rosmarinus officinalis*, *Myrtus communis*, *Origanum compactum*, and *Eugenia aromatica*. These plants are renowned for their diverse biological activities. Antioxidant activities were evaluated using DPPH, ABTS, and TAC tests. Antifungal activity was tested against four postharvest pathogens associated with chickpea in storage: *Fusarium culmorum*, *Rhizopus oryzae*, *Penicillium italicum*, and *Aspergillus niger*, using the broth microdilution technique. Additionally, the efficacy of several major compounds against fungi found in the EOs 1,8-cineole, carvacrol, and eugenol was evaluated. Furthermore, this study explored the potential synergy of combining eugenol and carvacrol in various ratios. Based on the results, *E. aromatica* EO exhibited the highest antioxidant activity, as evidenced by its lowest IC_50_ values for a DPPH of 0.006 mg/mL. This EO also demonstrated the best antifungal activity, with MIC values ranging from 0.098 to 0.13 μL/mL. The high concentration of eugenol in this oil was identified as a contributing factor to its potent antifungal effects. The individual application of eugenol displayed significant antifungal efficacy, which was further enhanced by incorporating carvacrol at a 1:3 ratio. This synergistic combination presents promising potential for the development of specific formulations aimed at optimizing grain protection during storage.

## 1. Introduction

The food industry is actively researching alternative preservation methods that are crucial for mitigating food waste and ensuring food security [1,2]. Seed storage is particularly vulnerable to oxidation and fungal infection during storage, resulting in significant quantity and quality losses. Free radical oxidative damage results in the degradation of lipids and proteins in seeds [3]. This oxidative process produces aldehydes or ketones, which cause irreversible alterations in the taste, flavor, color, and texture of food products, in addition to a decline in nutritional quality and seed viability [4,5]. Moreover, reactive radicals generated by oxidation can initiate a chain reaction in the human body, disrupting the structure and function of healthy cells [1,6]. Furthermore, fungal infection during post-harvest storage can result in substantial losses in seed quality and quantity. Grain storage fungi can produce a range of mycotoxins that adversely affect human health [3,7], including aflatoxins, trichothecenes, and malformin.

Exploring natural bioactive substances like essential oils (EOs) presents a promising approach for improving post-harvest quality without resorting to chemical methods, offering a sustainable and environmentally conscious alternative [4]. EOs are recognized for their wide array of biological activities, such as antioxidant, antibacterial, antiviral, insecticidal, and antifungal properties [5,8,9]. They offer potential substitutes for synthetic antioxidants such as butylated hydroxytoluene (BHT) and butylated hydroxyanisole (BHA), which are associated with significant health risks and toxicity [10,11]. Additionally, EOs exhibit potent antifungal activities against various storage fungi, and their relatively harmless, biodegradable, and non-toxic nature makes them widely accepted by consumers [12,13,14]. EOs consist of complex mixtures of volatile organic compounds, including terpenoids and phenylpropanoids. Terpenoids, classified as monoterpene, sesquiterpene, or diterpene, consist of two, three, or four isoprene units [15]. The major elements of EOs have undergone extensive investigation in the pharmaceutical and cosmetic sectors [16,17] as well as for use as flavorings and preservatives in the food industry [18,19]. The combination of these key compounds offers a novel approach to enhancing their antimicrobial and antioxidant activities [20].

The increasing need to explore natural products with versatile functions for food preservation is evident [21]. These compounds can simultaneously serve as preservatives, antioxidants, and antimicrobial agents, fulfilling the demand for clean-label and environmentally friendly food products for consumers [22]. Extracts derived from plants containing polyphenols and EOs exemplify these adaptable natural products, providing diverse advantages for improving food safety and quality [23]. Accepting these compounds will pave the way for a safer, healthier, and more sustainable food industry, aligning with evolving consumer preferences [24,25].

This study aims to develop effective formulations to optimize the protection of grain storage by (1) evaluating in vitro the antioxidant and antifungal activities of EOs extracted from four aromatic and medicinal plant species, *Rosmarinus officinalis*, *Myrtus communis*, *Origanum compactum*, and *Eugenia aromatica*; (2) investigating the antifungal activity of some major components of the EOs cineol, carvacrol, and eugenol against storage fungi; and (3) assessing the potential synergy of combining eugenol and carvacrol in varying ratios to optimize the antifungal activity.

## 2. Results

### 2.1. Essential Oil Yields

The evaluation of EO quantities in relation to dry plant matter revealed a significant range of yield variability among distinct plant species (Table 1). The yields ranged from 0.4% to 15%, highlighting the diversity among species. Notably, *E. aromatica* demonstrated the highest yield, while the lowest yield (0.4%) was observed for *M. communis*.

### 2.2. Chemical Composition of Essential Oils

The results of the chemical composition analysis of the EOs are concisely summarized in Table 2. The EOs displayed a complex combination of monoterpene hydrocarbons, oxygenated monoterpenes, and sesquiterpene hydrocarbons. In *R. officinalis,* a substantial percentage of 1,8-cineole (48.22%) was identified, along with a notable quantity of camphor (22.58%). Similarly, *M. communis* exhibited significant content of 1,8-cineole (42.22%), along with α-pinene (17.56%) and other compounds listed in Table 2. The EO of *E. aromatica* was characterized by the notable presence of eugenol (96.29%). Remarkably, *O. compactum* exhibited a high level of carvacrol (79.20%). These findings emphasize the distinct chemical profiles of each botanical species, distinguished by specific compounds that likely play essential roles in their biological activities. The major constituents of these EOs, namely 1,8-cineole, carvacrol, and eugenol (Figure 1), were chosen to investigate their antifungal effectiveness against post-harvest fungi.

### 2.3. Antioxidant Activity of Essential Oils

Among the four species tested, the EO of *E. aromatica* demonstrated significant antioxidant effectiveness, even surpassing that of ascorbic acid, as shown in Table 3. The IC_50_ values for the DPPH and ABTS assays were 0.006 mg/mL and 0.024 mg/mL, respectively. In contrast, *O. compactum*, *R. officinalis*, and *M. communis* exhibited relatively lower antioxidant activity (IC50 > 150 μg/mL) in both DPPH and ABTS assays. These results emphasize the notable ability of all analyzed EOs to scavenge DPPH and ABTS radicals. Additionally, the evaluation of reducing capacity further confirmed the strength of *E. aromatica*, demonstrating the highest reducing capacity, equivalent to 748.36 ascorbic acid units. In comparison, *O. compactum*, *R. officinalis*, and *M. communis* displayed moderate antioxidant capacity.

### 2.4. Antifungal Activity

#### 2.4.1. Antifungal Activity of Essential Oils

The antifungal activity of EOs extracted from four distinct plant species was assessed against four fungal strains (*F. culmorum*, *R. oryzae*, *P. italicum*, and *A. niger*) (Figure 2). Table 4 displays the values for the minimum inhibitory concentration (MIC) and the minimum fungicidal concentration (MFC). All tested extracts demonstrated inhibitory effects on the fungal strains. *E. aromatica* EO showed the highest antifungal effectiveness, with MIC values ranging from 0.098 to 0.13 μL/mL. Interestingly, *E. aromatica* EO performed better than the commercial fungicide (Azoxystrobin) against most of the tested fungi. However, it is important to note that *E. aromatica* EO showed a fungistatic effect, limiting fungal growth without complete eradication. *O. compactum* EO also showed significant efficacy, with MIC values ranging from 0.781 to 3.125 μL/mL and with a fungicidal effect. In contrast, the extracts of *R. officinalis* and *M. communis* showed relatively modest antifungal activity, characterized by MIC values ranging from 6.25 to 25 μL/mL, indicating reduced effectiveness against the tested fungi.

#### 2.4.2. Antifungal Activity of the Main Components of EOs

The antifungal activity of the main components of the tested EOs (1,8-cineole, carvacrol, and eugenol) was evaluated against the four post-harvest fungi (Table 5). Notably, carvacrol demonstrated very high antifungal activity, with MIC values ranging from 0.098 to 0.13 μL/mL. Moreover, fungicidal effects were observed at concentrations ranging from 0.195 to 0.39 μL/mL. Similarly, eugenol showed high antifungal activity, displaying MIC values ranging from 0.195 to 0.781 μL/mL, along with fungicidal attributes. In contrast, 1,8-cineole exhibited restricted antifungal potential, characterized by higher MIC values. All tested compounds exhibited fungicidal properties against all fungal strains.

#### 2.4.3. Antifungal Activity of the Main Component Mixture

The antifungal activity of binary mixtures of eugenol and carvacrol against four post-harvest fungi was investigated (Table 6). The results indicate that mixture 2, composed of 25% eugenol and 75% carvacrol (in a ratio of 1:3), demonstrates greater efficacy than the other mixtures. The MIC value for this mixture was determined to be 0.098 μL/mL against all four fungi. Mixture 1, containing an equal proportion of the two components, showed a slightly reduced level of effectiveness.

Binary combination 2 demonstrated an additive effect, displaying fractional inhibitory concentration index (FICI) values ranging from 0.69 to 0.81 (0.5 ≤ FICI ≤ 1), while the other mixtures showed indifferent effects (1 ≤ FICI ≤ 4) (Table 7). The combination of eugenol and carvacrol resulted in enhanced antifungal effectiveness against the tested fungi.

### 2.5. Correlation Analysis between Antifungal and Antioxidant Activities

Pearson’s correlation analysis revealed a positive correlation between the antioxidant activity, quantified through the DPPH and ABTS assays, and the antifungal activity against the fungal strains (*F. culmorum*, *R. oryzae*, *P. italicum*, and *A. niger*) (Figure 3). The correlation coefficients was ranged from R^2^ = 0.73 to R^2^ = 0.99. The ABTS assay, in particular, showed a strong correlation with all fungal MICs, with coefficient values ranging from 0.9 to 1. Furthermore, a negative correlation was observed between TAC and fungal MICs of fungi, with correlation coefficients ranging from −0.59 to −0.46.

## 3. Discussion

The composition of essential oils (EOs) varies significantly due to factors such as plant species, harvest location, climate, and extraction techniques used [26]. EOs, rich in diverse chemical components, exhibit a range of biological activities and modes of action [27].

Among the EOs investigated in this study, *E. aromatica* EO demonstrated the highest antioxidant activity. This antioxidant potency, characterized by a low IC_50_ value of 6 µg/mL (<50 µg/mL), aligns with the findings of Saptarini et al., 2020 [28]. *E. aromatica* EO not only displayed a high scavenging capacity but also exhibited a notable reduction ability. Conversely, EOs of *O. compactum*, *R. officinalis*, and *M. communis* exhibited comparatively lower antioxidant effects, with *R. officinalis* and *M. communis* showing similar activities. Previous studies have highlighted clove oil’s excellent antioxidant properties, which corroborate our findings [29,30,31,32]. The antioxidant activity of clove oil might stem from its major component, eugenol, along with the complex interplay among its constituents, potentially involving synergistic and antagonistic effects [33]. The prominent presence of eugenol, constituting about 96% of *E. aromatica* EO’s composition, contributes to its antioxidant activity. Eugenol’s neuroprotective properties and its ability to inhibit lipid peroxidation are well documented [34,35]. Its structural capacity to stabilize phenoxy radicals through hydrogen atoms and neutralize damaged molecules underscores its role in preventing oxidative damage and carcinogenic mutations [36,37]. Notably, Jirovetz et al. (2006) revealed that clove EO exhibits DPPH radical scavenging activity at lower doses than eugenol alone, implying potential synergies within the EO’s components [38]. Additionally, Dahham et al. (2015) showed that caryophyllene, which composes 2% of *E. aromatica,* possesses significant antioxidant activity [39]. Eugenyl acetate, accounting for 0.7% of the EO, exhibited even greater antioxidant activity than clove EO, with IC_50_ values of 367.5 μg/mL and 283.9 μg/mL, respectively [40]. Therefore, *E. aromatica* EO’s antioxidant potential can be partially attributed to eugenol and the synergistic interactions among its phenolic compounds and secondary metabolites. Interestingly, carvacrol, comprising around 79% of *O. compactum* EO, displayed important antioxidant activity when tested individually [41]. Extensive studies corroborate carvacrol’s potent antioxidant properties [42,43]. Notably, oregano EO, which contains carvacrol, showcased higher antioxidant activity than carvacrol itself [44]. This suggests a synergistic action among *O. compactum* EO’s constituents, with carvacrol being the key contributor. *M. communis* and *R. officinalis* EOs, both featuring 1,8-cineole as a major compound (approximately 48% and 42% of composition, respectively), exhibited similar antioxidant effects. The synergistic effects of 1,8 cineole have been documented [45]. Phenolic and flavonoid compounds often account for EOs’ antioxidant attributes, acting as electron donors in free radical reactions [46,47], as also indicated by Noshad et al. (2021) [48].

Our study demonstrated significant inhibition of fungal strains by the tested EOs. Notably, *E. aromatica* EO exhibited the highest antifungal activity, even surpassing the efficacy of the fungicide azoxystrobin. *O. compactum* EO also showed pronounced fungicidal activity. In contrast, *M. communis* and *R. officinalis* EOs displayed lower antifungal effects. Consistent with previous research, clove and oregano oils are well-established for their antifungal activities [49,50,51,52,53]. *E. aromatica* EO’s strong inhibitory effect can be attributed to its major component, eugenol, or its synergetic interactions. Eugenol alone demonstrated substantial antifungal activity against the post-harvest fungal strains, although less effectively than *E. aromatica* EO. Eugenol’s minimum inhibitory concentration ranged from 0.195 to 0.781 μL/mL, while *E. aromatica* EO’s MIC values ranged from 0.098 to 0.13 μL/mL. Interestingly, eugenol displayed superior fungicidal properties compared to clove oil, aligning with Schmidt et al.’s findings (2007) [54]. This suggests that other components present in *E. aromatica* EO could enhance its antifungal activity synergistically. Carvacrol, the dominant compound in *O. compactum* EO (about 79%), exhibited the highest efficacy against fungal growth, with MIC values ranging from 0.098 to 0.13 μL/mL. Carvacrol outperformed *O. compactum* EO itself, which displayed MIC values ranging from 0.781 to 3.125 μL/mL. Our findings align with Schlösser et al.’s (2018) research [55], indicating carvacrol’s stronger antifungal effect compared to oregano oil. Multiple studies have highlighted carvacrol’s efficacy against various fungi, with low MIC values ranging from 0.1 to 0.2 μL/mL [56,57]. Hence, carvacrol appears to be the primary contributor to *O. compactum* EO’s antifungal activity. *R. officinalis* and *M. communis* EOs demonstrated similar antioxidant activities due to the presence of 1,8-cineole, their major compound. However, 1,8-cineole exhibited lower antifungal activity compared to the oils themselves. Our findings are in accordance with Dammak et al. (2019) [58], suggesting that 1,8-cineole’s antifungal potential is relatively modest compared to *L. nobilis* and *L. dentate* EOs. This indicates that other components in these EOs synergistically enhance their antifungal properties [59].

Our investigation extended to binary mixtures of eugenol and carvacrol to enhance their antifungal efficacy. A 25% eugenol to 75% carvacrol ratio exhibited the most promising results among the tested mixtures, with a MIC value of 0.098 µL/mL against the evaluated post-harvest fungi. The synergistic combination of eugenol and carvacrol demonstrated an additive effect, enhancing the overall antifungal efficacy. Previous research has highlighted the significant inhibitory effects of binary mixtures, such as carvacrol and thymol, major components of thyme oil, against various fungi [60,61]. Eugenol and citral, major constituents of citronella and lemongrass oils, have also shown potent fungal growth inhibition that surpasses their individual effects [62]. Our results are consistent with those of Schlosser et al. (2018) [55], which highlighted the additive antifungal effect of carvacrol and eugenol on foodborne mold fungi. Properly balanced mixtures can be valuable for food applications as long as sensory qualities are maintained [55]. Synergy among natural substances enhances stability and bioactivity [63]. Combining different major compounds can optimize absorption or penetration into fungal cells, producing an impact greater than the sum of its parts. Phenolic compounds and aromatic aldehydes enhance antifungal activity due to steric hindrance [64]. According to Nieto et al. (2013) [9], Alkyl groups added to the benzene ring can also enhance antifungal activity. Notably, EOs’ antimicrobial activity depends on their most common compounds’ composition and concentration [24]. The synergistic action of EO constituents can disrupt fungal growth by affecting different targets or stages of the fungal life cycle, often damaging cell membranes and altering permeability, leading to leakage of intracellular contents [59,65,66]. Therefore, the combined effect of eugenol and carvacrol optimizes antifungal properties, targeting multiple aspects of fungal growth.

In this study, a strong correlation emerged between antioxidant and antifungal activities, suggesting a potential synergistic relationship. This suggests the possibility of a dual-function natural food preservative. Monoterpenes in EOs can act as pro-oxidants in fungal cells, disrupting cycles and accumulating reactive oxygen species (ROS) [67]. This can prevent microbial adherence and biofilm formation [68]. Hence, exploring antioxidants as fungal infection treatments is promising. *E. aromatica* EO, with its dual antifungal and antioxidant actions or its major constituent, eugenol, in combination with carvacrol, holds potential applications.

While this initial study has revealed significant antioxidant and antifungal activities present in used essential oils (EOs) and their primary components, as demonstrated by in vitro analyses, further exploration through more comprehensive in vivo investigations is necessary. These investigations will assist in uncovering the most optimal approaches for utilizing their potential to effectively protect chickpea seeds.

## 4. Materials and Methods

### 4.1. Plant Materials and Extraction of Essential Oils

Four species of aromatic and medicinal plants, representing two different plant families (*Lamiaceae* and *Myrtaceae*), were harvested during the flowering season from their natural habitats at various locations in Morocco (Table 8).

The essential oils (EOs) were extracted from the dried plant materials through hydrodistillation using a Clevenger apparatus following Guenther’s method [69]. In this process, 250 g of plant organs were placed in a 5000 mL flask filled with distilled water. The extraction proceeded for 3 h. This procedure was repeated iteratively to accumulate the required amount of EO for subsequent chemical and natural analyses. The extracted EOs were stored at 4 °C until use. The average EO yields were calculated based on dry plant material using Equation (1):(1)$$\mathrm{EO yield (\%)}=\frac{\mathrm{Weight of EO (g)}}{\mathrm{Weight of dried plant material (g)}}\times 100$$

### 4.2. Gas Chromatography–Mass Spectrometry Analysis of Essential Oil

The chemical composition of the tested EOs was analyzed using gas chromatography coupled with mass spectrometry (GC–MS) in Rabat, Morocco. The identification of the different constituents of the essential oil was carried out using PerkinElmer Clarus 580 GC/Mass Spectrometer. A capillary column (30 m × 0.25 mm; 0.25 μm film thickness) was utilized for the separation of different compounds of the essential oils. The carrier gas used was helium, with a flow rate of 1 mL/min. The temperature program was fixed from 50 to 230 at 4 °C/min, with a final hold time of 5 min. The injector and detector were maintained at 235 to 240 °C. The injection volume was 0.02 μL with splitless mode. The quantification of individual components of essential oils was performed using a PerkinElmer autosystem XL gas chromatograph equipped with a flame ionization detector (GC-FID). The column used was a capillary DB5 (0.25 μm film thickness and 30 m × 0.25 mm i.d.).

### 4.3. Major Components of the EOs

Essential oil major components were of the highest purity available. Eugenol (≥99%), carvacrol (≥97%), and cineole (≥98%) were purchased from Fluka Chemica. These compounds were utilized to assess the antifungal activity against fungi strains. As a positive control, the commercial fungicide Azoxystrobin (250 g/L), obtained from Syngenta, was used.

### 4.4. Antioxidant Activity

The evaluation of the antioxidant activity of the EOs was performed using two scavenging methods: DPPH (2,2-diphenyl-1-picrylhydrazyl) and ABTS (2,2′-azino-bis(3-ethylbenzothiazoline-6-sulfonic acid)). The determination of reducing capacity was carried out using the total antioxidant capacity (TAC) test. The ABTS and DPPH assays specifically target radicals, whereas the total antioxidant capacity assay offers a more comprehensive assessment of the overall antioxidant activity.

#### 4.4.1. DPPH Radical Scavenging Activity

The DPPH assay was employed to determine the IC_50_ values, which represent the concentration of each EO required for 50% inhibition of the free radical. The methanolic DPPH solution was prepared with a concentration of 0.004% *w*/*v* and allowed to stabilize for one hour. Subsequently, 100 μL of each methanolic EO solution at varying concentrations (*R. officinalis*: 10–200 mg/mL; *M. communis*: 10–200 mg/mL; *E. aromatica*: 0.01–0.05 mg/mL; *O. compactum*: 1–10 mg/mL) was mixed with 750 μL of freshly prepared DPPH. The samples were then incubated in the dark at room temperature for 30 min. The change in color from purple to yellow was observed to assess the radical scavenging capacity of the volatiles relative to the negative control (DPPH solution only). As a positive control, a methanolic solution of ascorbic acid was prepared with concentrations ranging from 0.001 to 0.05 mg/mL. The absorbance of the samples was measured at 517 nm using a UV spectrophotometer [70]. The test was carried out in triplicate, with each experiment performing three repetitions. The antioxidant activity was calculated using Equation (2):(2)$$\mathrm{Scavenging (\%)}=(\frac{\mathrm{AC}-\mathrm{AS}}{\mathrm{AC}})\times 100$$
where AC and AS represent the absorbencies of the negative control and samples.

The antioxidant activity of each sample was quantified using IC_50_ values, which indicate the concentration (in mg/mL) needed to inhibit DPPH radical formation by 50%. The IC_50_ values, along with their corresponding 95% confidence intervals, were derived through linear regression analysis of the dose–response curve. In this analysis, the percentage of inhibition was plotted against the concentration.

#### 4.4.2. ABTS Anion Radical Scavenging Activity

The ABTS solution was prepared by mixing 7 mM ABTS with 2.45 mM potassium persulfate in distilled water. The mixture was then incubated in the dark at room temperature for 16 h. The resultant ABTS+ solution was subsequently diluted with ethanol (1 mL per 1 mL) to attain an absorbance of 0.70 ± 0.02 at 734 nm [71]. For the experiment, 100 µL of each ethanolic solution of EO at varying concentrations was combined with 2 mL of fresh ABTS solution. The EO concentrations were as follows: *R. officinalis* (10–200 mg/mL), *M. communis* (10–200 mg/mL), *E. aromatica* (0.01–0.05 mg/mL), and *O. compactum* (1–8 mg/mL). After initial mixing, the absorbance was measured 6 min later at 734 nm. As a positive control, an ethanolic solution of ascorbic acid was prepared with concentrations ranging from 0.025 to 0.2 mg/mL. The test iwas conducted in three repetitions, and the entire experiment was repeated in triplicate. The percentage of inhibition was calculated using Equation (2), the same equation used for the DPPH-scavenging assay.

The antioxidant activity categories based on the IC_50_ values obtained from the scavenging activity of EOs are as follows: very strong when IC_50_ < 50 μg/mL, strong when 50 μg/mL < IC_50_ < 100 μg/mL, moderate effect when 100 μg/mL < IC_50_ < 150 μg/mL, and low when IC_50_ > 150 μg/mL [28].

#### 4.4.3. Total Antioxidant Capacity

The determination of total antioxidant capacity was accomplished via the phosphomolybdenum assay [72]. The reagent solution was prepared by dissolving 4 mM ammonium molybdate, 28 mM sodium phosphate, and 0.6 M sulfuric acid in distilled water. For the assay, 25 µL of each methanolic EO solution at a concentration of 1 mg/mL was mixed with 1 mL of the reagent solution. The samples were then incubated in a water bath at 95 °C for 90 min. The reduction of the Mo complex was indicated by a color change in the samples from transparent to green. The optical density of the samples was measured at 695 nm using a UV spectrophotometer against a blank. The test was conducted in three repetitions, and the entire experiment was performed in triplicate. The antioxidant activity was expressed in terms of ascorbic acid equivalents (mg AAE/g of extract).

### 4.5. Antifungal Activity

#### 4.5.1. Fungal Isolates and Inoculum Preparation

Four fungal species from distinct genera were employed in this assay: Fusarium (*F. culmorum*), Rhizopus (*R. oryzae*), Penicillium (*P. italicum*), and Aspergillus (*A. niger*). These post-harvest fungal species were isolated from chickpea seeds after storage. Identification of the isolates was conducted using standard identification keys based on their macroscopic and microscopic features [73,74,75]. These fungal species were chosen due to their prevalence within the seeds and their tendency to produce mycotoxins, such as zearalenone, B trichothecene, patulin, and ochratoxin A. The strains were maintained in a solution composed of 80% sabouraud dextrose broth (SDB) and 20% glycerol at −80 °C. Fungal strains were cultured on potato dextrose agar (PDA) for 7 to 10 days at a temperature of 22 ± 2 °C.

The inoculum suspension was prepared by rinsing the surface of the agar plates with 1 mL of sterile 0.9% saline water containing 0.1% Tween 20. Conidial suspensions were quantified using a hemocytometer and diluted to achieve a working suspension of 1.10^6^ spores/mL.

#### 4.5.2. Determination of the Minimum Inhibitory Concentration (MIC)

The antifungal activity of EOs and their main components was evaluated using the broth microdilution technique in sterile 96-well microplates [57]. The minimum inhibitory concentration (MIC) was defined as the lowest concentration of oil or main component that completely inhibited visible fungal growth. Initially, each well of the microplates received 150 µL of SDB. Subsequently, 150 µL of a double-concentrated emulsion of each test product was introduced to the wells of the first row of the plate. Doubling dilution series were executed to establish a concentration range from 0.004% to 5%, with preparations in dimethyl sulfoxide (DMSO) attaining a final concentration of 5% (*v*/*v*). Following this, 15 µL of fungal suspension inoculum was introduced to all wells of the microplate, leading to a final concentration of 1.10^5^ spores/mL. Wells containing inoculum but lacking treatment served as a negative control. A chemical treatment (azoxystrobin, 250 g/L) was utilized as the positive control. The plates were covered and incubated for 72 h at 22 ± 2 °C. Absorbance was measured at 492 nm using a microplate reader, with visual confirmation of outcomes under a microscope. Each test was replicated in triplicate, and the entire experiment was performed twice. The percentage of mycelium growth inhibition was computed using the subsequent Equation (3):(3)$$\mathrm{Mycelium growth inhibition (\%)}=\left[\frac{\left({AC}_{\left(72\right)}-{AC}_{\left(0\right)})-({AT}_{\left(72\right)}-{AT}_{\left(0\right)}\right)}{{(AC}_{\left(72\right)}-{AC}_{\left(0\right)})}\right]\times 100$$
where AC is the absorbance of the inoculum suspension in the control at t = 0 h and t = 72 h, and AT is the absorbance of the treatment at t = 0 h and t = 72 h.

#### 4.5.3. Determination of the Minimum Fungicidal Concentration (MFC)

The minimum fungicidal concentration (MFC) was determined to assess the fungicidal or fungistatic attributes of the EOs and their main components. MFC was defined as the lowest concentration leading to complete inhibition of mycelium growth on SDB plates, as indicated by subculturing wells showing the MIC. For MFC determination, 100 µL of contents from the MIC well and wells with concentrations surpassing the MIC were subcultured into a new microplate freshly supplemented with 150 µL of SDB [76]. These microplates were incubated under the same conditions as the MIC microplate, and growth was assessed visually through microscopy and absorbance measurement at 492 nm. The percentage of mycelium growth was calculated using Equation (3). Each test was executed in triplicate, with the experiment repeated twice to ensure accuracy and reproducibility. To determine the fungicidal or fungistatic impact of EOs, MFC/MIC ratios were computed. A ratio of MFC/MIC ≤ 4 denoted a fungicidal effect, while a ratio of MFC/MIC > 4 indicated a fungistatic effect [49].

#### 4.5.4. Determination of the Fractional Inhibitory Concentration Index (FICI)

The fractional inhibitory concentration (FIC) was determined to reveal the synergistic antifungal effect of two main components, carvacrol and eugenol. FIC determination was accomplished via the checkerboard method within a 96-well microtiter plate, adhering to the same procedure for determining the MIC. The FIC index was ascertained using the two-fold dilution method based on the results of the MIC determination for each main component separately. Concentrations of 4 × MIC, 2 × MIC, 1 × MIC, 1/2 × MIC, 1/4 × MIC, 1/8 × MIC, and 1/16 × MIC were chosen for the test [62]. Three binary combinations of eugenol and carvacrol were examined using distinct volume proportions (Table 9). The test was performed in three repetitions, and the experiment was conducted in duplicate.

The synergistic effect was expressed according to the following expression:FICI = Σ FIC = FIC (A) + FIC (B)

FIC (A) = MIC of the main compound (A) in combination/MIC of the main compound (A) alone.

FIC (B) = MIC of the main compound (B) in combination/MIC of the main compound (B) alone.

The values obtained for this index determined the combined effect of both compounds: FICI–FIC index values were interpreted as follows: FICI values ≤ 0.5 indicated synergy, 0.5 ≤ FICI ≤ 1 indicated an additive effect, 1 ≤ FICI ≤ 4 indicated an indifferent effect, and FICI values > 4 indicated an antagonistic effect [49].

### 4.6. Statistical Analysis

The results were presented as the mean ± standard deviation. Mean differences were assessed using analysis of variance (one-way ANOVA), followed by the Tukey post hoc test for multiple group comparisons. A significance level of *p* < 0.05 was considered to indicate statistical significance. The statistical analysis was performed using IBM SPSS v.22 software. Pearson’s correlation test was utilized to establish correlation coefficients between antifungal activity and antioxidant activity (DPPH, ABTS, and TAC) using Origin 2023 (10.0) software.

## 5. Conclusions

In conclusion, our study underscores the significant antioxidant and antifungal potential of *E. aromatica* among the tested EOs. The substantial presence of eugenol in this oil can contribute to its pronounced antifungal properties. Furthermore, the application of eugenol alone exhibited a notable antifungal effect, which was further potentiated when combined with carvacrol in a 1:3 ratio. This synergistic interaction suggests the possibility of developing customized formulations or treatments to enhance grain protection during storage. The combination of *E. aromatica* EO and carvacrol also presents a promising prospect for advancing natural preservation in diverse industries. Further investigation is required to optimize this combination and identify optimal application strategies.

## Figures and Tables

**Figure 1 plants-12-03587-f001:**
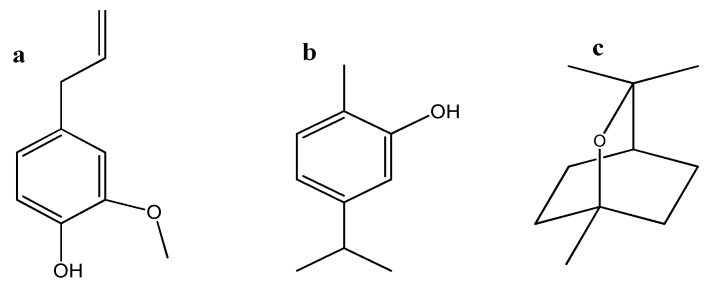
Molecular structures of three monoterpenes using ChemDraw (version 12.0). (**a**) Eugenol; (**b**) carvacrol; (**c**) 1,8-Cineole.

**Figure 2 plants-12-03587-f002:**
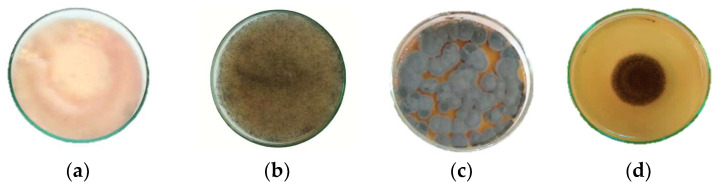
Cultures of selected fungal species on potato dextrose agar: (**a**) *Fusarium culmorum*, (**b**) *Rhizopus oryzae*, (**c**) *Penicillium italicum*, (**d**) *Aspergillus niger*.

**Figure 3 plants-12-03587-f003:**
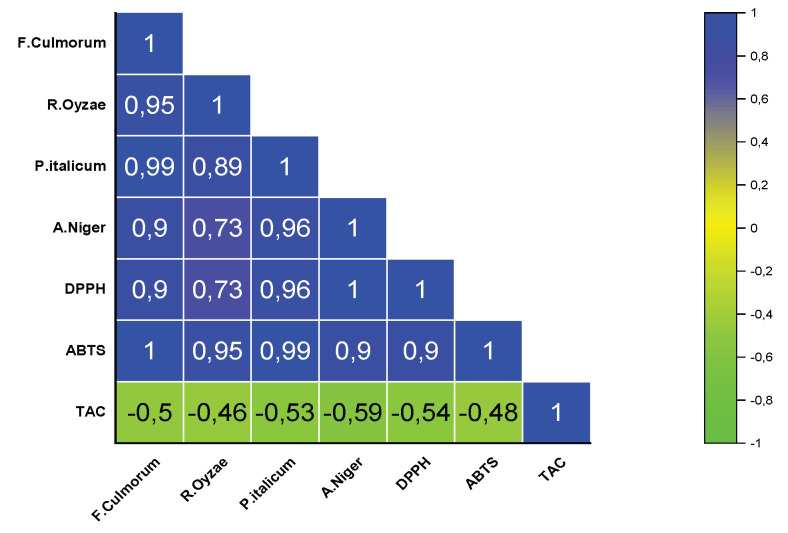
Pearson’s correlation coefficient of the antifungal and antioxidant activities of EOs. *Fusarium culmorum, Rhizopus oryzae, Penicillium italicum, Aspergillus niger*; DPPH = 2.2−diphenyl−1−picrylhydrazyl; ABTS = 2.2’−azino−bis(3−ethylbenzothiazoline−6−sulfonic acid); TAC = total antioxidant capacity.

**Table 1 plants-12-03587-t001:** EO yields extracted from four species of aromatic and medicinal plants.

Essential Oils	Extraction Yield (%)
*Rosmarinus officinalis*	2.52
*Origanum compactum*	1.6
*Eugenia aromatica*	15
*Myrtus communis*	0.4

**Table 2 plants-12-03587-t002:** Relative percentages of the main components identified in EOs.

Essential Oils	Compounds	Molecular Formula	Composition (%)
*Rosmarinus officinalis*	α-Pinene	C_10_H_16_	2.45 ^a^
	Camphene	C_10_H_16_	3.77 ^c^
	β-Myrcene	C_10_H_16_	2.80 ^b^
	1,8-Cineole	C_10_H_18_O	48.22 ^f^
	Camphor	C_10_H_16_O	22.58 ^e^
	α-Terpineol	C_10_H_18_O	4.36 ^d^
*Myrtus communis*	α-Pinene	C_10_H_16_	17.56 ^f^
	1,8-Cineole	C_10_H_18_O	42.22 ^g^
	Linalool	C_10_H_18_O	2.80 ^b^
	α-Terpineol	C_10_H_18_O	3.31 ^c^
	Myrtenol	C_10_H_16_O	3.32 ^d^
	Myrtenyl acetate	C_12_H_18_O_2_	15.06 ^e^
	Geranyl acetate	C_12_H_20_O_2_	1.92 ^a^
*Eugenia aromatica*	Eugenol	C_10_H_12_O_2_	96.29 ^d^
	Eugenyl acetate	C_12_H_14_O_3_	0.72 ^b^
	α-Humulene	C_15_H_24_	0.54 ^a^
	Caryophyllene	C_15_H_24_	2.02 ^c^
*Origanum compactum*	α-Thujene	C_10_H_16_	2.26 ^c^
	Myrcene	C_10_H_16_	4.85 ^e^
	p-Cymene	C_10_H_14_	3.91 ^d^
	γ-Terpinene	C_10_H_16_	0.33 ^a^
	Carvacrol	C_10_H_14_O	79.20 ^f^
	Caryophyllene	C_15_H_24_	1.55 ^b^

The letters represent significant differences between the compounds of EOs, determined by one-way ANOVA followed by Tukey’s post hoc comparison test, with a significance level of *p* < 0.05.

**Table 3 plants-12-03587-t003:** Evaluation of antioxidant activity of EOs using free radical DPPH, ABTS, and TAC assays.

Essential Oils	Reducing Capacity TAC (mgAE/gEO)	Scavenging Capacity	
		DPPH IC_50_ (mg/mL)	ABTS IC_50_ (mg/mL)
*R. officinalis* *O. compactum* *E. aromatica* *M. communis*	199.952 ± 7.393 ^b^ 154.238 ± 12.957 ^a^ 748.365 ± 3.016 ^c^ 200.429 ± 3.898 ^b^	91.117 ± 3.180 ^b^ 3.937 ± 0.111 ^a^ 0.006 ± 0.001 ^a^ 91.312 ±1.146 ^b^	48.940 ± 2.484 ^b^ 1.813 ± 0.042 ^a^ 0.0241 ± 0.0001 ^a^ 98.501 ± 1.105 ^c^
Ascorbic acid	-	0.037 ± 0.003 ^a^	0.132 ± 0.000 ^a^

IC50 = Half maximal inhibitory concentration. The presented results are expressed as mean values ± standard deviation (*n* = 3). The letters represent significant differences between the oils for each column. The test of comparison was determined by the Student–Newman–Keuls test at a significance level of *p* < 0.05.

**Table 4 plants-12-03587-t004:** Evaluation of MIC and MFC of EOs against four fungi species.

Essential Oils		*F. culmorum*	*R. oryzae*	*P. italicum*	*A. niger*
*R. officinalis*	MIC (µL/mL) MFC (µL/mL) MFC/MIC Interpretation	6.25 ^c^ 6.25 ^b^ 1.00 Fungicide	6.25 ^d^ 25.00 ^c^ 4.00 Fungicide	8.33 ± 2.08 ^b^ 25.00 ^c^ 3.00 Fungicide	12.50 ^c^ 50.00 ^c^ 4.00 Fungicide
*O. compactum*	MIC (µL/mL) MFC (µL/mL) MFC/MIC Interpretation	0.781 ^b^ 1.302 ± 0.260 ^a^ 1.67 Fungicide	3.125 ^c^ 6.25 ^b^ 2.00 Fungicide	0.781 ^a^ 3.125 ^b^ 4.00 Fungicide	1.562 ^b^ 3.125 ^b^ 2.00 Fungicide
*E. aromatica*	MIC (µL/mL) MFC (µL/mL) MFC/MIC Interpretation MIC (µL/mL) MFC (µL/mL) MFC/MIC Interpretation	0.13 ± 0.03 ^a^ 0.651 ± 0.130 ^a^ 4.99 Fungistatic 12.50 ^d^ 25.00 ^c^ 2.00 Fungicide	0.098 ^a^ 0.781 ^a^ 7.97 Fungistatic 25.00 ^e^ 50.00 ^d^ 2.00 Fungicide	0.13 ± 0.03 ^a^ 1.562 ^a^ 11.98 Fungistatic 12.50 ^b^ 50.00 ^d^ 4.00 Fungicide	0.098 ^a^ 0.781 ^a^ 7.97 Fungistatic 12.50 ^c^ 50.00 ^c^ 4.00 Fungicide
*M. communis*					
Azoxystrobin	MIC (µL/mL)	1.302 ± 0.451 ^b^	0.520 ± 0.226 ^b^	0.651 ± 0.226 ^a^	0.098 ^a^

MIC = minimum inhibitory concentration; MFC = minimum fungicidal concentration. The presented results are expressed as mean values ± standard deviation (*n* = 3). The letters indicate significant differences between the oils for each fungus, determined by a one-way ANOVA followed by a Tukey post hoc comparison test, with a significance level of *p* < 0.05.

**Table 5 plants-12-03587-t005:** Evaluation of the MIC and MFC of the main components of EOs against four fungi.

Main Components		*F. culmorum*	*R. oryzae*	*P. italicum*	*A. niger*
1,8-Cineole	MIC (µL/mL) MFC (µL/mL) MFC/MIC Interpretation	12.50 ^c^ 25.00 ^c^ 2.00 Fungicide	10.437 ^c^ 41.75 ^c^ 4.00 Fungicide	17.39 ± 3.47 ^b^ 83.5 ^c^ 4.00 Fungicide	41.75 ^c^ 167.00 ^c^ 4.00 Fungicide
Carvacrol	MIC (µL/mL) MFC (µL/mL) MFC/MIC Interpretation	0.098 ^a^ 0.195 ^a^ 1.99 Fungicide	0.098 ^a^ 0.195 ^a^ 1.99 Fungicide	0.13 ± 0.03 ^a^ 0.39 ^a^ 3.00 Fungicide	0.098 ^a^ 0.195 ^a^ 1.99 Fungicide
Eugenol	MIC (µL/mL) MFC (µL/mL) MFC/MIC Interpretation	0.39 ^b^ 0.781 ^b^ 2.00 Fungicide	0.39 ^b^ 1.562 ^b^ 4.01 Fungicide	0.195 ^a^ 0.781 ^b^ 4.01 Fungicide	0.781 ^b^ 1.562 ^b^ 2.00 Fungicide

MIC = minimum inhibitory concentration; MFC = minimum fungicidal concentration. The presented results are expressed as mean values ± standard deviation (*n* = 3). The letters indicate significant differences between the oils for each fungus, determined by a one-way ANOVA followed by a Tukey post hoc comparison test with a significance level of *p* < 0.05.

**Table 6 plants-12-03587-t006:** Evaluation of MIC and MFC of the three mixtures of carvacrol and eugenol against four fungi.

Mixtures		*F. culmorum*	*R. oryzae*	*P. italicum*	*A. niger*
Mix. 1	MIC (µL/mL) MFC (µL/mL)	0.260 ± 0.113 ^a,b^ 0.39 ^a^	0.195 ^b^ 0.781 ^b^	0.163 ± 0.056 ^a,b^ 0.651 ± 0.226 ^a^	0.260 ± 0.113 ^a,b^ 0.39 ^a^
Mix. 2	MIC (µL/mL) MFC (µL/mL)	0.098 ^a^ 0.39 ^a^	0.098 ^a^ 0.39 ^a^	0.098 ^a^ 0.651 ± 0.226 ^a^	0.098 ^a^ 0.325 ± 0.113 ^a^
Mix. 3	MIC (µL/mL) MFC (µL/mL)	0.39 ^b^ 0.39 ^a^	0.39 ^c^ 0.781 ^b^	0.195 ^b^ 1.562 ^b^	0.39 ^b^ 0.781 ^b^

MIC = minimum inhibitory concentration; MFC = minimum fungicidal concentration; Mix. 1 = 1:1 ratio; Mix. 2 = 1:3 ratio; Mix. 3 = 3:1 ratio. The presented results are expressed as mean values ± standard deviation (*n* = 3). The letters indicate significant differences between the oils for each fungus, determined by a one-way ANOVA followed by a Tukey post hoc comparison test with a significance level of *p* < 0.05.

**Table 7 plants-12-03587-t007:** Evaluation of the synergistic antifungal effect of eugenol–carvacrol mixtures.

Mixtures		*F. culmorum*	*R. oryzae*	*P. italicum*	*A. niger*
Carvacrol (µL/mL)	MIC	0.098	0.098	0.13 ± 0.03	0.098
Eugenol (µL/mL)	MIC	0.39	0.39	0.195	0.781
Mix. 1	MIC of Mix 1 FIC of CAR FIC of EUG FICI	0.260 ± 0.113 1.33 0.33 1.66 Indifferent	0.195 0.99 0.25 1.24 Indifferent	0.163 ± 0.056 0.62 0.42 1.04 Indifferent	0.260 ± 0.113 1.33 0.17 1.49 Indifferent
Mix. 2	MIC of Mix 2 FIC of CAR FIC of EUG FICI Interpretation	0.098 0.75 0.06 0.81 Additive	0.098 0.75 0.06 0.81 Additive	0.098 0.57 0.13 0.69 Additive	0.098 0.75 0.03 0.78 Additive
Mix. 3	MIC of Mix 3 FIC of CAR FIC of EUG FICI Interpretation	0.39 0.99 0.75 1.74 Indifferent	0.39 0.99 0.75 1.74 Indifferent	0.195 0.38 0.75 1.13 Indifferent	0.39 0.99 0.37 1.37 Indifferent

FICI = fractional inhibitory concentration index; FIC = fractional inhibitory concentration; CAR = carvacrol; EUG = eugenol; Mix. 1 = 1:1 ratio; Mix. 2 = 1:3 ratio; Mix. 3 = 3:1 ratio. The results are presented as mean values ± standard deviation *(n* = 3).

**Table 8 plants-12-03587-t008:** Aromatic and medicinal plant species investigated in this study.

Species Name	Family	Parts Used	Harvesting Location	Harvesting Season
*Rosmarinus officinalis*	*Lamiaceae*	Leaves + small stems	Midelt region	June–September
*Origanum compactum*	*Lamiaceae*	Leaves + small stems	Ouazzane region	June–August
*Myrtus communis*	*Myrtaceae*	Leaves	Marrakech region	May–June
*Eugenia aromatica*	*Myrtaceae*	Flowers	Herbalist	-

**Table 9 plants-12-03587-t009:** The three binary mixture ratios of eugenol and carvacrol used.

Mixture	Eugenol	Carvacrol
Mix. 1	1	1
Mix. 2	1	3
Mix. 3	3	1

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
