# Peer review of "In Vitro Antioxidant and Antifungal Activities of Four Essential Oils and Their Major Compounds against Post-Harvest Fungi Associated with Chickpea in Storage"

_plants, 2023, doi:10.3390/plants12203587_

Round 1
Reviewer 1 Report
In this manuscript (plants-2575684) entitled "In vitro Antioxidant and Antifungal Activities of Four Essential Oils and their major compounds for Preserving Grain Quality during Storage" submitted to Plants, Lamyae ET-TAZY and colleagues have assesses the antifungal and antioxidant properties of essential oils (EOs) from four plants: Rosmarinus officinalis, Myrtus communis, Origanum compactum, and Eugenia aromatica. This research is interesting, but the present manuscript is unsuitable for publication.
1. Growth pictures of four fungal species used in this assay: Fusarium (F. culmorum), Rhizopus (R. oryzae), Penicillium (P. italicum), and Aspergillus (A. niger) should be shown in the revised manuscript.
2. For Table 4, analysis in significance of difference should be performed. Please label the significance of difference in the revised column of composition (%).
3. For Tables 6, 7, 8 and 9, data presented here should be carefully checked in the revision. For instance, ±0.00 and ±0.000 have appeared many times, which is abnormal. In addition, the comma and dot presented in numbers were incorrectly used.
4. For Figure 2, Full names of abbreviations F.Culmurum, R.Oyzae, P.italicum, A.Niger, DPPH, ABTS and TAC presented in the figure should be described at the revised legend.
Moderate editing of English language required
Author Response
We greatly value your manuscript review and the insightful comments you've provided, which have notably enhanced the quality of our work. We have implemented numerous improvements to the manuscript, including enhancing its English language quality to ensure a clear and coherent presentation of our results. Your thoughtful recommendations have been meticulously integrated, all visibly marked in yellow.
Please find a detailed point-by-point response to your comments:
Point 1: Growth pictures of four fungal species used in this assay: Fusarium (F. culmorum), Rhizopus (R. oryzae), Penicillium (P. italicum), and Aspergillus (A. niger) should be shown in the revised manuscript.
Response 1: Thank you for your suggestion. We have included pictures of the fungal species used in the revised manuscript. These images were obtained from cultures of fungi grown on potato dextrose agar, which was used in this assay (Lines 180-181).
Point 2: For Table 4, analysis in significance of difference should be performed. Please label the significance of difference in the revised column of composition (%).
Response 2: Done. We have conducted a statistical analysis for the data presented in Table 4 and have labeled the significant differences in the composition (%) column.
Point 3: For Tables 6, 7, 8 and 9, data presented here should be carefully checked in the revision. For instance, ±0.00 and ±0.000 have appeared many times, which is abnormal. In addition, the comma and dot presented in numbers were incorrectly used.
Response 3: Done. The data presented in Tables 6, 7, 8, and 9 has been thoroughly reviewed. Additionally, we have addressed any inconsistencies in the use of commas and dots in numbers. Thank you for bringing these issues to our attention.
Point 4: For Figure 2, Full names of abbreviations F.Culmoum, R.Oyzae, P.italicum, A.Niger, DPPH, ABTS and TAC presented in the figure should be described at the revised legend.
Response 4: Done. Lines 379-381.
Point 5: Moderate editing of English language required
Response 5: Done. We have conscientiously performed revisions to improve the clarity and coherence of the language across the entire document. Notably, all modifications have been distinctly highlighted (in green) throughout the text.
Point 6: All the cited references relevant to the research can be improved
Response 6: Done. We have meticulously examined the list of references in our manuscript and have taken care to incorporate seven recent citations to enhance its scholarly relevance:
- Soković, M.D; Vukojević, J.; Marin, P.D; Brkić, D.D.; Vajs, V.; van Griensven L.J.L.D. Chemical Composition of Essential Oils of Thymus and Mentha Species and Their Antifungal Activities. Molecules, 2009, 14, 238–249. https://doi.org/10.3390/molecules14010238
- Tresserra-Rimbau, A.; Lamuela-Raventos, R.M.; Moreno, J.J. Polyphenols, food and pharma. Current knowledge and directions for future research. Pharmacol., 2018, 156, 186–195. https://doi.org/10.1016/j.bcp.2018.07.050
- Hernandez Ochoa, L.R. Substitution de solvants et matières actives de synthèse par un combiné «solvant/actif » d’origine végé Doctoral dissertation, 2005.
- Hyldgaard, M.; Mygind, T.; Meyer, R.L. Essential Oils in Food Preservation: Mode of Action, Synergies, and Interactions with Food Matrix Components. Microbiol., 2012, vol. 3. https://doi.org/10.3389/fmicb.2012.00012
- Langeveld, W.T.; Veldhuizen, E.J.A.; Burt, S.A. Synergy between essential oil components and antibiotics: a review. Crit. Microbiol., 2014, 40, 76–94. https://doi.org/10.3109/1040841X.2013.763219
- Ong, K.S.; Mawang, C.I.; Daniel-Jambun, D.; Lim, Y.Y.; Lee, S.M. Current anti-biofilm strategies and potential of antioxidants in biofilm control. Expert Rev. Anti-Infect. Ther., 2018, 16, 855–864. https://doi.org/10.1080/14787210.2018.1535898
- Rhimi, W.; Theelen, B.; Boekhout, T.; Otranto, D.; Cafarchia, C. Malassezia spp. Yeasts of Emerging Concern in Fungemia. cell. infect. microbiol., 2020, 10, 370. https://doi.org/10.3389/fcimb.2020.00370
Reviewer 2 Report
The present document evaluates the potential antifungal and antioxidanteffect of 4 essential oils.
The introduction is well written and provides a good overview and the objectives are clear
The material and methods section should be partially completed
There are few minor things to correct
In the material and method section
You need to add, the country and the city of the GC, you need to add the brad of the column, if the injection was split (and the rate) or splitless and, the quantity injected
How the main compounds found in the EO were identified?
Line 98: replace to post-harvest
Replace IC50 by IC50 : all along the document
Table 5, please indicate in the footnote that the statistical analysis was performed in columns
Table 9, there is no letter indicating the statistical significance, whereas it’s noted in the footnote. It will facilitate the lecture of the table to put in the legend which is the composition of each mixture analysed
Line 355 a dot starts the sentence, please correct
Line 385, correct English
The results are clearly presented and the conclusion is complete.
The conclusion is clear.
My concern goes to the title, as the only link to the preservation of the seeds is stated in the introduction, but latter there is no relationship with how the tested EO can be used to protect those seeds, as fighting against the fungus in vitro as weel as an antioxidant fonction do not guarantee the protection of the seeds that were not used in the analysis. How you can link them? Maybe changing a little the title, or open the conclusion or the discussion
Few english errors can be foung
Author Response
We are grateful for your comprehensive review of our manuscript and the perceptive comments you provided, which have profoundly enriched the quality of our work. Our material and methods section has been meticulously enhanced to ensure its comprehensiveness and adequacy. We have thoughtfully integrated your suggestions, leading to substantial revisions across the manuscript, all meticulously highlighted in yellow.
Please find a detailed point-by-point response to your comments:
Point 1: In the material and method section, you need to add, the country and the city of the GC, you need to add the brad of the column, if the injection was split (and the rate) or splitless and, the quantity injected. How the main compounds found in the EO were identified?
Response 1: Done. Thank you for your valuable input. We did the clarifications of all this recommendations in the section of ‘’gas chromatography-mass spectrometry analysis of essential oil’’. (Lines 87-99).
Point 2: Line 98: replace to post-harvest
Response 2: Done.
Point 3: Replace IC50 by IC50 : all along the document
Response 3: Done.
Point 4: Table 5, please indicate in the footnote that the statistical analysis was performed in columns
Response 4: Done. (Lines 300-301).
Point 5: Table 9, there is no letter indicating the statistical significance, whereas it’s noted in the footnote, It will facilitate the lecture of the table to put in the legend which is the composition of each mixture analysed
Response 5: Done. thank you for your remark . The irrelevant footnote in Table 9 has been removed. Additionally, we have enhanced clarity by providing a detailed description of the composition of each mixture in the legend of the table 8 and 9, as you recommended. (Line 364).
Point 6: Line 355 a dot starts the sentence, please correct
Response 6: Done.
Point 7: Line 385, correct English
Response 7: Done.
Point 8: My concern goes to the title, as the only link to the preservation of the seeds is stated in the introduction, but latter there is no relationship with how the tested EO can be used to protect those seeds, as fighting against the fungus in vitro as well as an antioxidant function do not guarantee the protection of the seeds that were not used in the analysis. How you can link them? Maybe changing a little the title, or open the conclusion or the discussion.
Response 8: We appreciate your concern about the link between the tested essential oil and seed preservation. We have modified a little the title and expanded the discussion section to provide a clearer connection. (Lines 479-486).
Point 9: Few english errors can be found
Response 9: Thank you for your observation. We have carefully reviewed the manuscript to address the English errors, ensuring its overall quality and clarity (in green).